# Efficient Knowledge Distillation from Model Checkpoints

**Chaofei Wang**\*, **Qisen Yang**\*, **Rui Huang, Shiji Song, Gao Huang**†
Department of Automation, Tsinghua University, China
`wangcf18, yangqs19, hr20@mails.tsinghua.edu.cn`
`shijis, gaohuang@tsinghua.edu.cn`

## Abstract

Knowledge distillation is an effective approach to learn compact models (students) with the supervision of large and strong models (teachers). As empirically there exists a strong correlation between the performance of teacher and student models, it is commonly believed that a high performing teacher is preferred. Consequently, practitioners tend to use a well trained network or an ensemble of them as the teacher. In this paper, we observe that an intermediate model, *i.e.*, a checkpoint in the middle of the training procedure, often serves as a better teacher compared to the fully converged model, although the former has much lower accuracy. More surprisingly, a weak snapshot ensemble of several intermediate models from a same training trajectory can outperform a strong ensemble of independently trained and fully converged models, when they are used as teachers. We show that this phenomenon can be partially explained by the information bottleneck principle: the feature representations of intermediate models can have higher mutual information regarding the input, and thus contain more "dark knowledge" for effective distillation. We further propose an optimal intermediate teacher selection algorithm based on maximizing the total task-related mutual information. Experiments verify its effectiveness and applicability. Our code is available at https://github.com/LeapLabTHU/CheckpointKD.

## 1 Introduction

Knowledge distillation (KD) [1, 2] has been proved to be an effective technique to promote the performance of a low-capacity model by transferring "dark knowledge" from a large teacher model. Empirically, there usually exists a strong correlation between the performance of the teacher model and the student model. For this reason, it is a standard practice to use a well trained network or an ensemble of multiple well trained networks as the teacher [3, 4, 5], and some researches are attempted to improve distillation performance via boosting the ensemble performance [6, 7]. The underlying assumption is that high performing teachers lead to better student models.

However, this viewpoint has been challenged by some recent works [8, 9, 10, 11, 12], in which it has been observed that a large model capacity gap between the teacher and student may have a negative effect for knowledge transfer. To address this issue, researchers have proposed to employ an intermediate-size network [8] or an assistant network [9] to improve the distillation performance in such scenarios. In [10], a "tolerant" teacher model is designed by using a softened loss function. In [11], Park et al. proposed to learn student-friendly teacher by plugging in student branches during the training procedure. Nevertheless, there is no clear theoretical explanation for the gap between teacher and student, and the search for a substitute teacher is not straightforward.

---

\*Equal contribution
†Corresponding author

36th Conference on Neural Information Processing Systems (NeurIPS 2022).

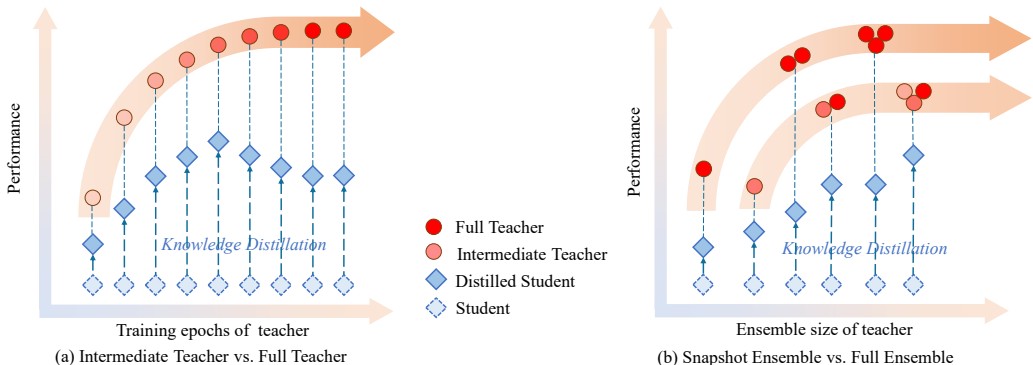

Figure 1: A sketch map of the two counterintuitive observations. (a) A weak intermediate model can serve as a better teacher than the strong fully converged model. (b) A weak Snapshot Ensemble [13] can serve as a better teacher than the strong Full Ensemble.

In this paper, we make an intriguing observation that further supports the viewpoint that *high performing models may not necessarily be good teachers*, but from a novel perspective. Specifically, we find that an unconverged intermediate model from the middle of the training procedure, often serves as a better teacher than the final converged model, although the former has much lower accuracy (as illustrated in Figure 1(a)) . Moreover, a weak snapshot ensemble of intermediate teacher models along the same optimization path (denoted as Snapshot Ensemble, which is a variant of [13][3]) can outperform the standard ensemble of an equal number of independently trained teacher models (denoted as Full Ensemble). This surprising phenomenon is illustrated in Figure 1(b), in which a Snapshot Ensemble can have better distillation performance than a Full Ensemble, although the accuracy of the former is significantly lower.

To understand the above phenomenon, we show that there is a strong connection between KD and the information bottleneck (IB) theory [14]. Therein, it has been observed that during the training procedure of deep neural networks, the mutual information between the learned features $F$ and the target $Y$, denoted as $I(Y; F)$, increases monotonically as a function of the training epochs; while the mutual information between $F$ and the input $X$, denoted as $I(X; F)$, grows in the early training stage, but then decreases gradually after a certain number of epochs. We note that maximizing the mutual information $I(Y; F)$ is helpful for improving the teacher model itself, but not always necessary for KD, because the ground truth target $Y$ is already included in the KD objective function. In contrast, the mutual information $I(X; F)$, to some extent, can be viewed as a type of dark knowledge that is desired for effective KD. For example, considering an image with a man driving a car, although it may be uniquely labeled into the "car" category, it still contains features of the "people" category. Such weak but non-negligible features extracted from the input (measured by $I(X; F)$) are in fact the most valuable knowledge for distilling student models. Not surprisingly, most KD algorithms apply a high temperature to soften the network prediction in order to reveal these information from a teacher model. However, as shown by IB theory, a fully converged model tends to be overconfident and may already have collapsed representations for non-targeted classes. Therefore, simply scaling the temperature can not effectively recover the suppressed knowledge. On the contrary, an intermediate model, although does not reach its top accuracy due to non-optimal $I(Y; F)$, may have a larger $I(X; F)$ that benefits KD. This partially explains our observation that intermediate models can be better teachers. More detailed and formal analyses are provided in the following sections.

Further, we propose an optimal intermediate teacher selection algorithm based on the IB theory. From the perspective of entropy, the teacher model's representation can be decomposed as the information with respect to the input, the output and some nuisance [15]. The proposed algorithm aims to find the most informative intermediate teacher model which possesses the minimal part of nuisance on a training trajectory. Experiments verify its applicability in various distillation scenarios. Our contributions are summarized as follows:

- By designing two exploratory experiments, we observe the phenomenon that intermediate models can serve as better teachers than fully converged models. This suggests that for

---

[3]In this paper,we adopt a normal cosine learning rate instead of the cyclic learning rate.

effective KD one should not only focus on improving the teacher performance. Instead, rethinking what the "dark knowledge" is and how to enrich it is highly valuable.

- We demonstrate the connection between our observations and the IB theory, providing a new perspective for understanding KD and explaining the "dark knowledge".
- Based on our observations and analyses, a novel simple but effective algorithm is proposed to find the optimal intermediate teacher and achieve better distillation performance. Experiments validate its effectiveness and adaptability.

## 2 Related Work

**Knowledge distillation.** Hinton et al. [2] proposed to transfer "dark knowledge" of a strong capacity teacher model to a compact student model by minimizing the Kullback-Leibler divergence between the soft targets of the two models. Since then, many variants of KD methods were proposed to improve the distillation performance[16], such as Fitnets [17], AT[18], CCM[19], FSP[20], SP[21], CCKD[22],TC$^3$KD[23]. As a promising technique to improve model generalization, ensemble learning is often combined with knowledge distillation to improve the distillation performance[3, 11, 4]. In the online knowledge distillation framework[24], efforts were made to boost the distillation performance by increasing the diversity between multiple models to improve the ensemble performance [6, 7]. Most existing methods commonly assumed that a high performing teacher is preferred for KD. On the contrary, some researchers thought that the model capacity gap between strong teachers and small students usually degrades knowledge transfer[8, 9, 12, 25]. Some of them have experimentally verified that poor teachers can also perform KD tasks well[12, 25]. Several methods are proposed to compress this gap by introducing an assistant network [8, 9] or designing a student-friendly teacher [10, 11]. However, they did not explain theoretically why gap exists and how gap affects KD. In the self-distillation framework[26, 27], it is essentially the intermediate model that is used as the teacher, but no one has a theoretical explanation for why the intermediate model works. In this paper, we link KD and IB theory through extensive observations and experiments. From the perspective of mutual information, we explain why intermediate models serve as better teachers than full models, and how to select an suitable intermediate model to reduce the negative impact of model gap.

**Information bottleneck.** Tishby et al. [28] firstly proposed the information bottleneck concept and provided a tabular method to numerically solve the IB Lagrangian (Eq. (3)). Later, Tishby and Zaslavsky [14] proposed to interpret deep learning with IB principle. Following this idea, Shwartz-Ziv and Tishby [29] studied the IB principle to explain the training dynamics of deep networks. It has motivated many studies to apply the IB principle to interpret and improve deep neural networks (DNNs) [30, 31, 32]. Recently, some researchers introduced IB principle to deep reinforcement learning successfully [33, 34, 35]. As far as we know, we are the first to introduce IB principle to interpret knowledge distillation.

## 3 Exploratory Experiments

In this section, we first formally describe the KD and ensemble KD methods used in the paper, then design two exploratory experiments to show how intermediate models are surprisingly valuable for KD, despite their lower accuracies due to incompletion of training.

### 3.1 Formulation

In the classical KD setting, a fully converged teacher model (full teacher for short) $T^{\text{full}}$ is used to distill a student model $S$. Define $P_{T^{\text{full}}}$ as the softmax output of teacher model, $P_S$ as the softmax output of student model and $Y_{\text{true}}$ as the true labels. The student model is trained to optimize the following loss function:

$$L_{\text{KD}} = \alpha H(Y_{\text{true}}, P_S) + (1 - \alpha) H(P_{T^{\text{full}}}^\tau, P_S^\tau), \tag{1}$$

where $H$ refers to the cross-entropy, $\alpha$ is the trade-off parameter, $\tau$ is the temperature. Conducting KD with an intermediate teacher model $T^{\text{inter}}$, means using $T^{\text{inter}}$ instead of $T^{\text{full}}$ in Eq. (1).

In the standard ensemble KD setting, there are $M(M \geq 2)$ full teachers $\left\{T_1^{\text{full}}, T_2^{\text{full}}, ..., T_M^{\text{full}}\right\}$, which have the same network structure and training strategy but different initial parameters. The student

model needs to mimic the average softened softmax output of all teacher models. We call this method Full Ensemble KD. The loss function is as follows:

$$L_{\text{EKD}} = \alpha H(Y_{\text{true}}, P_{\text{S}}) + (1 - \alpha) H(\frac{1}{M} \sum_{i=1}^{M} P_{\text{T}_i^{\text{full}}}^{\tau}, P_{\text{S}}^{\tau}). \tag{2}$$

The Snapshot Ensemble aggregates $M(M \geq 2)$ intermediate teachers $\{T_1^{\text{inter}}, T_2^{\text{inter}}, ..., T_M^{\text{inter}}\}$ from one training trajectory. Conducting KD with a Snapshot Ensemble, means using $T_i^{\text{inter}}$ instead of $T_i^{\text{full}}$ in Eq. (2). We call it Snapshot Ensemble KD.

## 3.2 Experimental design and setups

To examine the common assumption *"high performing teachers lead to better student models"* and explore the value of intermediate models, we design two experiments. 1) The standard KD is to train a full teacher model to distill a student model. What if we adopt the intermediate teacher models instead? 2) The standard ensemble KD is to train multiple full teacher models independently and average their output to distill a single student model. What if we adopt the Snapshot Ensemble instead of the Full Ensemble? We name the first experiment as "Intermediate Teacher vs. Full Teacher", and the second experiment as "Snapshot Ensemble vs. Full Ensemble". For generality, we conduct experiments on the CIFAR-100 [36], Tiny-ImageNet[37] and ImageNet [38] datasets with various teacher-student pairs. The distillation loss functions follow Eqs. (1) and (2). For fair comparison, we search the optimal hyperparameters (*i.e.*, the loss ratio $\alpha$ and the temperature $\tau$ ) for each teacher-student pair. Top 1 accuracy is averagely evaluated in five independent experiments.

The "Intermediate Teacher vs. Full Teacher" experiment is conducted on the CIFAR-100 and ImageNet. On CIFAR-100, we adopt WRN-40-2[39] and ResNet-110[40] as teacher models, WRN-40-1[39], ResNet-32[40], and MobileNetV2[41] (width multiplier is 0.75) as student models. We train each teacher model for 200 epochs to ensure convergence. We save the intermediate models at the $20^{\text{th}}$, $40^{\text{th}}$, ..., $180^{\text{th}}$ epochs as intermediate teachers, and the models at the $200^{\text{th}}$ epoch as full teachers. On ImageNet, we adopt ResNet-50 [40], and ResNet-34 [40] as teacher models, and MobileNetV2 [41], and ResNet-18 [40] as student models. We follow the standard PyTorch practice but train teacher models for 120 epochs to guarantee convergence. We save the intermediate models at the $60^{\text{th}}$ epoch as intermediate teachers, and the models at the $120^{\text{th}}$ epoch as full teachers. The "Snapshot Ensemble vs. Full Ensemble" experiment is conducted on CIFAR-100 and Tiny-ImageNet. We train models for 150 epochs on Tiny-ImageNet to ensure convergence. We save the intermediate models at the $75^{\text{th}}$ epoch as intermediate teachers, and the models at the $150^{\text{th}}$ epoch as full teachers. We adopt WRN-40-1[39], ResNet-32[40], and MobilenetV2[41] as student models, and WRN-40-2[39], and ResNet-110[40] as teacher models. Due to the page limitation, we include the introduction of the datasets and detailed experimental settings in the Appendix A.1.

## 3.3 Intermediate Teacher vs. Full Teacher

Firstly, we simply compare the half-way teachers with the full teachers on CIFAR-100 and ImageNet. It means that the intermediate models at the $100^{\text{th}}$ epoch are adopted as $T^{\text{inter}}$ on CIFAR-100, the intermediate models at the $60^{\text{th}}$ epoch are adopted as $T^{\text{inter}}$ on ImageNet. The training cost of all intermediate teachers is only half that of the full teachers. Table 1 shows the comparison results. Specifically, on CIFAR-100, for WRN-40-2, the accuracy of the intermediate model is 13.54% lower than that of the full model, but its distillation performance is comparable (0.08% higher) and superior (0.96% higher). For ResNet-110, the accuracy of the intermediate model is 13.98% lower than that of the full model, but its distillation performance is still comparable (0.01% higher and 0.16% higher). On ImageNet, the accuracy of the inter-

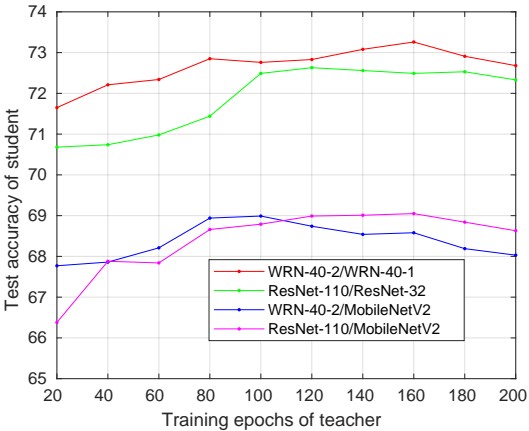

Figure 2: Ablation experiments of the training epochs of $T^{\text{inter}}$ on CIFAR-100.

Table 1: Comparison results of "Full Teacher vs. Intermediate Teacher". $T^{\text{full}}$ represents a full teacher. $T^{\text{inter}}$ represents a half-way intermediate teacher whose KD performance is underlined. $T^{\text{inter}}_*$ represents the best intermediate teacher whose KD performance is **bold-faced**. The numbers in brackets represent decreases ($\downarrow$) or increases ($\uparrow$) in accuracy.

| Dataset | Network structure | | Accuracy of reference models | | | Accuracy of KD | | |
|---|---|---|---|---|---|---|---|---|
| | T | S | $T^{\text{full}}$ | $T^{\text{inter}}$ | S | $T^{\text{full}}$ | $T^{\text{inter}}$ | $T^{\text{inter}}_*$ |
| CIFAR-100 | WRN-40-2 | WRN-40-1 | 76.53 | 62.99 ($\downarrow$ 13.54) | 70.38 | 72.68±0.10 | 72.76±0.24 ($\uparrow$ 0.08) | **73.26±0.03 ($\uparrow$ 0.58)** |
| | | MobileNetV2 | | | 64.49 | 68.03±0.34 | 68.99±0.12 ($\uparrow$ 0.96) | **68.99±0.12 ($\uparrow$ 0.96)** |
| | ResNet-110 | ResNet-32 | 73.41 | 59.43 ($\downarrow$ 13.98) | 70.16 | 72.48±0.22 | 72.49±0.32 ($\uparrow$ 0.01) | **72.63±0.13 ($\uparrow$ 0.15)** |
| | | MobileNetV2 | | | 64.49 | 68.63±0.35 | 68.79±0.17 ($\uparrow$ 0.16) | **69.05±0.27 ($\uparrow$ 0.42)** |
| ImageNet | ResNet-34 | ResNet-18 | 73.86 | 66.08 ($\downarrow$ 7.78) | 69.75 | 70.66 | 70.86 ($\uparrow$ 0.20) | **70.98 ($\uparrow$ 0.32)** |
| | ResNet-50 | MobileNetV2 | 76.93 | 69.14 ($\downarrow$ 7.79) | 64.18 | 66.64 | 66.79 ($\uparrow$ 0.15) | **66.92 ($\uparrow$ 0.28)** |

Table 2: Comparison results of "Full Ensemble vs. Snapshot Ensemble". $\text{EN}_1$ denotes $T_1^{\text{full}}+T_2^{\text{full}}$. $\text{EN}_2$ denotes $T_1^{\text{inter}}+T_2^{\text{full}}$. $\text{EN}_3$ denotes $T_1^{\text{inter}}+T_1^{\text{full}}$. The best results of ensemble KD performance are **bold-faced**, while the best results of ensemble performance are underlined.

| Dataset | Network structure | | Accuracy of baseline | | | Accuracy of ensemble KD | | | Accuracy of ensemble | | |
|---|---|---|---|---|---|---|---|---|---|---|---|
| | T | S | T | S | KD | $\text{EN}_1$ | $\text{EN}_2$ | $\text{EN}_3$ | $\text{EN}_1$ | $\text{EN}_2$ | $\text{EN}_3$ |
| CIFAR-100 | WRN-40-2 | WRN-40-1 | 76.53 | 70.38 | 72.68±0.10 | 73.05±0.16 | **73.80±0.31** | 73.70±0.04 | 79.44 | 76.90 | 76.29 |
| | | MobileNetV2 | | 64.49 | 68.03±0.34 | 68.69±0.32 | 69.14±0.27 | **69.20±0.31** | | | |
| | ResNet-110 | ResNet-32 | 73.41 | 70.16 | 72.48±0.22 | 72.88±0.14 | 72.91±0.17 | **73.03±0.24** | 76.92 | 74.28 | 73.23 |
| | | MobileNetV2 | | 64.49 | 68.63±0.35 | 69.69±0.25 | **70.46±0.34** | 70.19±0.25 | | | |
| Tiny-ImageNet | WRN-40-2 | WRN-40-1 | 57.61 | 53.85 | 55.65 | 55.96 | 56.27 | **56.37** | 62.81 | 61.53 | 60.33 |
| | | MobileNetV2 | | 53.75 | 56.53 | 56.67 | 57.05 | **57.28** | | | |
| | Average | | 79.32 | 74.11 | 76.04 | 66.11 | 66.61 | **66.63** | 73.06 | 70.90 | 69.95 |

mediate teachers is about 7.8% lower than that of the full teachers, but their distillation performance is still comparable or even better.

To further explore the potential of the intermediate models, we conduct ablation experiments of the training epochs of $T^{\text{inter}}$ on CIFAR-100. Figure 2 shows some valuable information. 1) Good and bad teachers both improve the baseline of student model, which is consistent with [25]. 2) The peaks of all curves are not the last points, which means that there is always an intermediate teacher that is better than the full teacher. Table 1 (the last column) shows the KD performance of the best $T^{\text{inter}}$. 3) Generally, the accuracy curve of student first rises then decreases along with the teacher's training epochs. Combing Table 1 and Figure 2, we find a counterintuitive observation:

**Observation 1.** *The distillation performance of an intermediate teacher model can be comparable with or even better than that of the fully converged teacher model, although the accuracy and training cost of the former is significantly lower.*

### 3.4 Snapshot Ensemble vs. Full Ensemble

First, we fix the ensemble size to 2 to avoid introducing other factors. We train two full teacher models, $T_1^{\text{full}}$ and $T_2^{\text{full}}$, with 200 epochs independently, and save their intermediate models, $T_1^{\text{inter}}$ and $T_2^{\text{inter}}$, at the $100^{\text{th}}$ epoch. For the Full Ensemble, we construct an ensemble with $T_1^{\text{full}}$ and $T_2^{\text{full}}$. For the Snapshot Ensemble, we build an ensemble with one full teacher model $T_1^{\text{full}}$ and its intermediate model $T_1^{\text{inter}}$. We use $T_1^{\text{full}} + T_2^{\text{full}}$ and $T_1^{\text{inter}} + T_1^{\text{full}}$ to represent the Full Ensemble and the Snapshot Ensemble, respectively. To further explore the intermediate and full models, we add an additional evaluation object, an ensemble with one intermediate model $T_1^{\text{inter}}$ and one extra full teacher model $T_2^{\text{full}}$ from another training trajectory. Such combination is represented as $T_1^{\text{inter}} + T_2^{\text{full}}$.

Table 2 shows two observations: 1) For the ensemble performance, $T_1^{\text{full}} + T_2^{\text{full}} > T_1^{\text{inter}} + T_2^{\text{full}} > T_1^{\text{full}} + T_1^{\text{inter}}$ is consistently true; 2) For the distillation performance, $T_1^{\text{inter}} + T_1^{\text{full}}$ and $T_1^{\text{inter}} + T_2^{\text{full}}$

have similar performance, but significantly outperform $T_1^{\text{full}} + T_2^{\text{full}}$. In average, the accuracy of Full Ensembles is 3.11% higher than that of Snapshot Ensembles, but the distillation accuracy of the former is 0.52% lower the latter. Comprehensively considering training cost and distillation performance, the Snapshot Ensemble ($T_1^{\text{full}} + T_1^{\text{inter}}$) is the best choice.

To further verify this cognition, we conduct an ablation experiment of the ensemble size $k$ on CIFAR-100. We adopt WRN-40-2/WRN-40-1 as the teacher-student pair. The Full Ensemble is composed of $k$ full teacher models, while the Snapshot Ensemble is composed of 1 full teacher model and its $k - 1$ intermediate models. For simplicity, the training process of a full teacher model is averagely divided into $k$ phases to obtain the $k - 1$ intermediate models. We evaluate the test accuracy of distilled students and the total training cost, which consists of training teachers and distilling the student. Figure 3 shows two phenomena. 1) The distillation performance of the Snapshot Ensemble is significantly better than that of the Full Ensemble, with the same ensemble size $k$. 2) The training cost of the Snapshot Ensemble is significantly lower than that of the Full Ensemble, with the same ensemble size $k$. Hence, Snapshot Ensembles can be more economical and efficient in the KD setting. Combining Table 2 and Figure 3, we obtain the second counterintuitive conclusion:

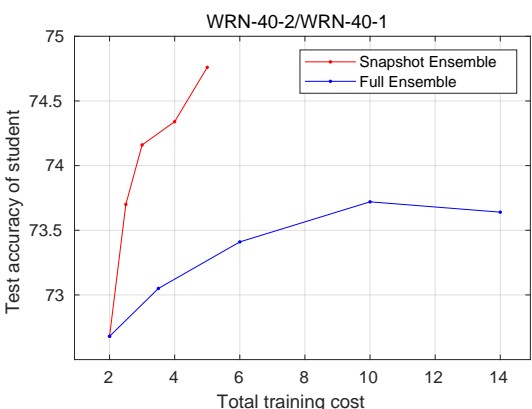

Figure 3: An ablation experiment of ensemble size ($k = 1, 2, 4, 7, 10$) on CIFAR-100. We estimate the training hours on TITAN Xp.

**Observation 2.** *A stronger ensemble does not always lead to a better distillation. In particular, the Snapshot Ensemble has worse ensemble performance and lower training cost but better distillation performance than the Full Ensemble.*

## 4 Knowledge Distillation and Information Bottleneck

We have observed that the student's accuracy first increases and then decreases while the teacher's accuracy increases monotonically, as a function of the training epoch of the teacher (as shown in Figure 2). In this section, we aim to understand this phenomenon using the IB principle proposed in [14]. The effect of mutual information on distillation performance is analyzed theoretically and experimentally, and further verified with class correlation information.

### 4.1 Connection knowledge distillation with information bottleneck

Tishby and Zaslavsky [14] proposed that layered neural networks form a Markov chain of successive representations of the input, and explained the optimization goal of DNNs with the IB principle. They claimed that DNNs tend to obtain an efficient representation of the input, capturing the features relevant to the output and compressing those irrelevant. Formally, for a DNN, define the input variable $X$ and desired output variable $Y$. Any representation of the input $F$, is defined through an encoder $P(F|X)$, and a decoder $P(Y|F)$. The optimization goal of the network can be described as the following IB trade-off optimization problem [29]:

$$\min_{F} \left\{ I(X; F) - \beta I(F; Y) \right\}, \tag{3}$$

where $I(X; F)$ represents the mutual information between $X$ and $F$, $I(F; Y)$ represents the mutual information between $F$ and $Y$, $\beta$ is a positive trade-off parameter. From a perspective of information theory, knowledge transfer can be expressed as retaining high mutual information between the teacher and student networks (proposed in VID [42]). Following Eq. (3), we define the representation of the teacher model $F_t$, the representation of the student model $F_s$. The optimization goal of the student model in KD setting can be described as follows:

$$\min_{s} \left\{ I(X; F_s) - \beta I(Y; F_s) - \gamma I(F_t; F_s) \right\}, \tag{4}$$

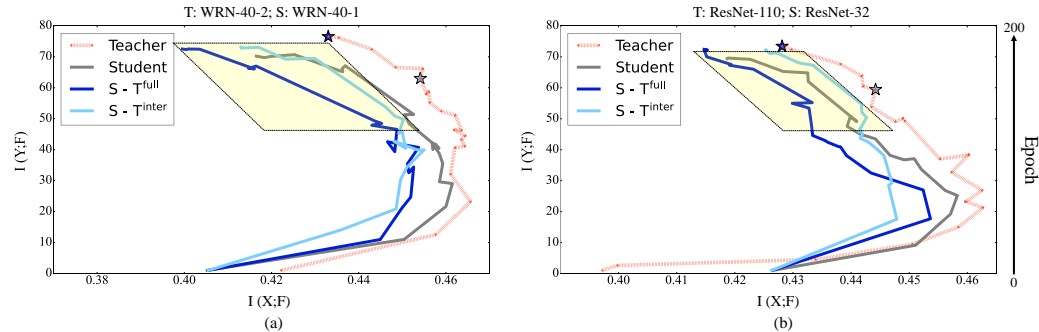

Figure 4: The mutual information curves on the information plane. The red lines represent teacher models. The dark blue $\star$ represents $T^{\text{full}}$ while the light blue $\star$ represents $T^{\text{inter}}$. The gray, dark blue and light blue lines represent students, distilled students with $T^{\text{full}}$ and distilled students with $T^{\text{inter}}$. The yellow areas are the observation areas. A full teacher compresses $I(X; F)$ of student, while an intermediate teacher slows down the compression (a) or even amplifies $I(X; F)$ of student (b).

where $\gamma$ is a positive trade-off parameter. $I(F_t; F_s)$ represents the mutual information between the teacher and the student. $F_t$ and $F_s$ can be quantified by their coordinates: $(I(X; F_t), I(Y; F_t))$ and $(I(X; F_s), I(Y; F_s))$ on the information plane. Hence, maximizing $I(F_t; F_s)$ can evolve into minimizing $|I(X; F_t) - I(X; F_s)| + |I(Y; F_t) - I(Y; F_s)|$. Then, we can reformulate Eq. (4) to

$$\min_s \{ I(X; F_s) - \beta I(Y; F_s) + \gamma |I(X; F_t) - I(X; F_s)| $$
$$+ \gamma |I(Y; F_t) - I(Y; F_s)| \}, \tag{5}$$

where how to remove the absolute value signs depends on the relative location of the coordinates $(I(X; F_t), I(Y; F_t))$ and $(I(X; F_s), I(Y; F_s))$ on the mutual information plane.

## 4.2 Mutual Information Analysis

In order to explore the role of mutual information in knowledge distillation, we quantitatively analyze the mutual information curves of different teacher-student pairs. In practice, following [43], we adopt the test accuracy to quantify $I(Y; F)$ and a reconstruction loss to quantify $I(X; F)$. To get the reconstruction loss, we connect a decoder following the last convolution layer of the network model to reconstruct the input $X$. We adopt WRN-40-2/WRN-40-1 and ResNet110/ResNet32 pairs on CIFAR-100. We train each teacher model for 200 epochs to get a full model $T^{\text{full}}$ and save the intermediate model $T^{\text{inter}}$ at the $100^{th}$ epoch. Detailed settings are given in Appendix B.1. The mutual information curves of teacher model, student model, distilled student models with $T^{\text{full}}$ and $T^{\text{inter}}$ are shown in Figure 4. Some important observations and inferences are summarized.

1) The trend of all curves is similar to that shown in [29] and consistent with IB principle [14]. That is, $I(X; F)$ first goes up and then goes down while $I(Y; F)$ goes up monotonically. It can be interpreted as: for $I(X; F)$, networks first absorb information of the input $X$ and later eliminate part of the information irrelevant to the target output $Y$; for $I(Y; F)$, networks continuously accumulate information relevant to the target output $Y$.

2) The curve of a large teacher model is usually on the right of the curve of a small student model. It means a large model generally has greater ability of information representation.

3) The full model $T^{\text{full}}$ compresses more $I(X; F)$ but retains more $I(Y; F)$ than the intermediate model $T^{\text{inter}}$. That is to say, a full model may discard more information of non-target classes from $X$.

4) The student models distilled with $T^{\text{full}}$ further compress $I(X; F)$ (the dark blue curves are on the left of the gray curves in Figure 4), while those distilled with $T^{\text{inter}}$ slow down the compression of $I(X; F)$ (the light blue curve in Figure 4(a)) or even amplify $I(X; F)$ (the light blue curve in Figure 4 (b)). Retaining more $I(X; F)$ seems to be a key factor to get a competitive performance from $T^{\text{inter}}$.

For a full teacher with a large $I(Y; F_t)$ but a small $I(X; F_t)$, Eq. (5) is reformulated to

$$\min_s \{ (1 + \gamma) I(X; F_s) - (\beta + \gamma) I(Y; F_s) \}. \tag{6}$$

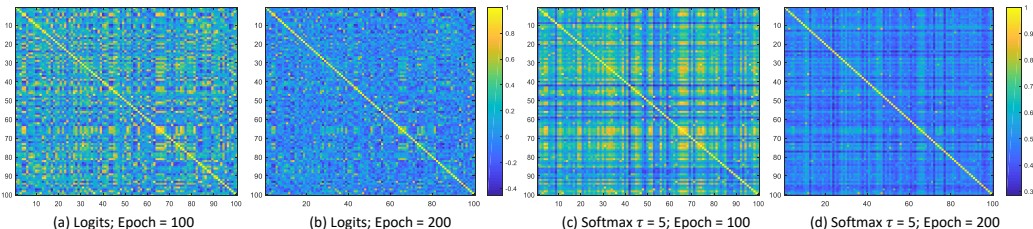

(a) Logits; Epoch = 100    (b) Logits; Epoch = 200    (c) Softmax $\tau$ = 5; Epoch = 100    (d) Softmax $\tau$ = 5; Epoch = 200

Figure 5: The heatmaps of class correlation information on the intermediate model (at the 100[th] epoch) and the full model (at the 200[th] epoch) on the CIFAR-100 dataset. (a) and (b) are calculated with the logits output; (c) and (d) are calculated with the softened softmax output (temperature is 5). More brightly colored areas mean that the intermediate model has more diversity than the full model.

For a suitable intermediate teacher model with a large $I(Y; F_t)$ and a large $I(X; F_t)$, Eq. (5) is reformulated to

$$\min_s \left\{ (1 - \gamma)I(X; F_s) - (\beta + \gamma)I(Y; F_s) \right\}. \tag{7}$$

Comparing Eq. (6) and Eq. (7), the difference is that a full teacher model accelerates the compression of $I(X; F_s)$ while the intermediate model alleviates the compression of $I(X; F_s)$. This is consistent with our $4^{th}$ observation. Therefore, we can reasonably assume that: *more mutual information with the input data can be the main reason why an intermediate teacher model achieves better distillation performance than a full teacher model.*

Compared with a full model, a suitable intermediate model has more $I(X; F)$ and less $I(Y; F)$, which means that the intermediate model has more non-target classes information. To do further analysis, we use heatmaps of cosine similarity to visualize the class correlation information contained in the intermediate model and the full model. For CIFAR-100, a heatmap is a $100 \times 100$ matrix $A$, where each element $A_{ij}$ represents the similarity between corresponding two classes. Specifically, we represent a class $c_i$ by its average logits or softmax output on the test set (a 100-dimensional vector $V_{c_i}$) and then calculate the cosine similarity $A_{ij}$ between $c_i$ and $c_j$ with the following equation:

$$A_{ij} = \frac{\langle V_{c_i}, V_{c_j} \rangle}{\|V_{c_i}\|\|V_{c_j}\|}. \tag{8}$$

Figure 5 shows some heatmaps of cosine similarity on the intermediate model and the full model. The network structure is WRN-40-2. Comparing Figure 5(a) with (b), we can see that the logits output of the intermediate model has better diversity than that of the full model. Comparing Figure 5(c) with (d), while using a softened softmax output with a higher temperature can increase some of the diversity between classes, the intermediate model still shows a wider range of class correlation information than the full model. Results of the class correlation information further support that a suitable intermediate teacher has more non-target classes information than a full teacher, which is consistent with the mutual information analysis.

### 4.3 Label smoothing regularization and knowledge distillation

The previous work [25] proposed that label smoothing regularization (LSR) can be considered as an ad-hoc KD with a pre-defined uniform distribution teacher. Mathematically for LSR, a uniform distribution $u$ is used in place of $P_{\text{Tfull}}^{\tau}$ in Eq. (1). We attempt to explore the connection between LSR and KD from the IB perspective. Obviously, the uniform distribution $u$ has a small $I(Y; F_t)$ and a small $I(X; F_t)$. Eq. (5) is reformulated to

$$\min_s \left\{ (1 + \gamma)I(X; F_s) - (\beta - \gamma)I(Y; F_s) \right\}, \tag{9}$$

where $\beta - \gamma$ must be positive to keep the optimization direction correct. Comparing Eq. (6) and Eq. (9), if we fix the coefficient of the second term $I(Y; F_s)$ to be 1, then the coefficient of the first term is $(1 + \gamma)/(\beta + \gamma)$ in Eq. (6) and $(1 + \gamma)/(\beta - \gamma)$ in Eq. (9). It means that LSR accelerates the compression of $I(X; F_s)$. In order to verify it visually, keeping the same setting as Figure 4, we add LSR to the student model and show its mutual information curve on the information plane. As shown in Figure 6, both the dark blue curve (the normal KD) and the yellow curve (LSR) are on the left of the gray curve (the student model). It means that both the normal KD and LSR play a similar role to

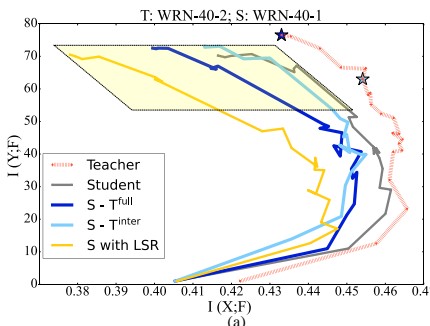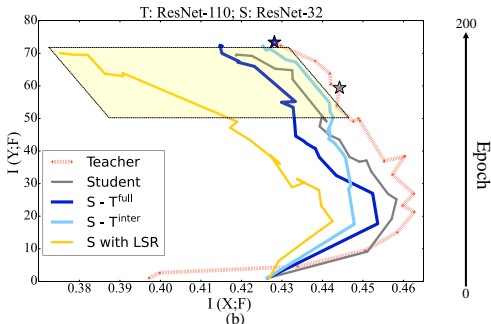

Figure 6: The mutual information curves on the information plane. The red curves represent the teacher model. The gray, dark blue and light blue curves represent the students, distilled students with full teacher and distilled students with intermediate teacher. The yellow curves represent student training with LSR. The yellow areas are the observation areas.

compress the student model's mutual information with respect to input, which supports the view in [25]. However, there is a big gap between the dark blue curve and the yellow curve, which means that the normal KD transfers more relevant information with input to the student model than the LSR does. For the distillation performance, $T^{\text{inter}} > T^{\text{full}} > u$ is consistent with the amount of $I(X; F_t)$ of the teacher models, which further supports our view in Sec. 4.2.

## 5 Optimal Intermediate Model Selection

As shown in Figure 2 and Figure 4, how to capture the optimal intermediate model is a non-trivial problem. An empirical conclusion is that a good intermediate model needs delicate trade-off between $I(X; F)$ and $I(Y; F)$. From the point of information entropy, the representation $F$ of model consists of information concerning the input $X$, the output $Y$ and nuisance $Z$ which is irrelevant to the task [15]. A formal description of the entropy of $F$ is:

$$H(F) = I(X; F) + I(Y; F) + Z. \quad (10)$$

To reduce the nuisance $Z$ and get a more informative intermediate teacher model, we solve the following optimization problem:

$$\max_{F} \left\{ I(X; F) + I(Y; F) \right\}, \quad (11)$$

where $F$ belongs to the set of representations in intermediate teacher models. Figure 7 is an example of the selecting process. The optimal solution is usually located in the upper right corner of the information plane. Based on the selecting strategy, we propose an algorithm to find the optimal intermediate teacher for effec-

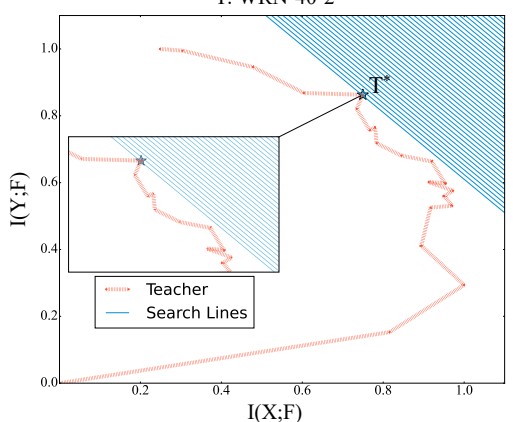

Figure 7: An example of selecting the optimal intermediate teacher $T^{\text{inter}}$. The red line represents the normalized mutual information curve of the teacher model (WRN-40-2) . The blue lines represent the selecting process.The star $\star$ represents the optimal intermediate teacher $T^*$ whose representation maximizes $\{I(X; F) + I(Y; F)\}$.

tive distillation as shown in Algorithm 1. Note that the value of mutual information $I(X; F)$ and $I(Y; F)$ is normalized to ensure the scale matching. Obtaining the optimal intermediate teacher model, standard KD or other mainstream KD methods can be conducted as usual.

To verify the superiority of the optimal intermediate model selection strategy, we conduct comparison experiments on the CIFAR-100 dataset. We keep the same experimental settings as in Sec. 3.3, but save teacher's checkpoints every 10 epochs. For each checkpoint, the information theoretic quantities $I(X; F)$ and $I(Y; F)$ are estimated as in Sec. 4.2. Then, we normalize $I(X; F)$ and $I(Y; F)$, and draw the mutual information curves. By algorithm 1, the optimal intermediate teacher models $T^*$ of WRN-40-2 and ResNet-110 are selected at the 160th and 120th epochs respectively. An

---

**Algorithm 1** Distillation with the optimal intermediate teacher.

---
1: Train a teacher model $T$ from scratch and save its checkpoint $T_i$ every M epochs;
2: Calculating the normalized $I(X; F)$ and $I(Y; F)$ of each $T_i$;
3: Draw the mutual information curve of $T$;
4: Find the optimal $T^*$ which has the representation that maximizes $\{I(X; F) + I(Y; F)\}$;
5: Take $T^*$ as the teacher model to do distillation.

---

Table 3: KD Results of the optimal intermediate models on CIFAR-100. The intermediate teacher models are selected at different epochs. The best results are **bold-faced**.

| Network structure | | Accuracy of T&S | | KD accuracy of different intermediate teachers | | | | |
|---|---|---|---|---|---|---|---|---|
| T | S | T | S | $T^{0.3}$ | $T^{0.5}$ | $T^{0.7}$ | $T^{\text{full}}$ | $T^*$ |
| WRN-40-2 | WRN-40-1 | 76.53 | 70.38 | 72.34±0.10 | 72.76±0.24 | 73.08±0.05 | 72.68±0.10 | **73.26**±0.03 |
| | MobileNetV2 | | 64.49 | 68.21±0.33 | **68.99**±0.12 | 68.54±0.07 | 68.03±0.34 | 68.58±0.34 |
| ResNet-110 | ResNet-32 | 73.41 | 70.16 | 70.74±0.18 | 72.49±0.32 | 72.46±0.30 | 72.48±0.22 | **72.63**±0.13 |
| | MobileNetV2 | | 64.49 | 67.84±0.26 | 68.79±0.17 | **69.01**±0.20 | 68.63±0.35 | 68.99±0.33 |
| Average | | 74.97 | 67.38 | 69.78 | 70.76 | 70.77 | 70.46 | **70.87** |

empirical conclusion is that the optimal intermediate model for each teacher model may be different. To evaluate the distillation performance of $T^*$, we pick out models at the 60th, 100th, 140th, 200th epoch, which are denoted by $T^{0.3}$, $T^{0.5}$, $T^{0.7}$ and $T^{\text{full}}$ as baselines. We adopt the same standard KD method [2] but different intermediate models as teachers. Table 3 shows that $T^*$ has the best average performance, which validates the effectiveness and adaptability of our selection strategy. In addition, $T^{0.7}$ and $T^{0.5}$ also have better average performance than $T^{\text{full}}$. If you think that selecting the optimal intermediate model is tedious, the half-way checkpoint may be your prefer.

## 6 Conclusion and Limitations

In this paper, we made an observation that an intermediate model can have richer "dark knowledge" than a fully converged model, and employed the IB principle to partially interpret this phenomenon. We argue that over-training of the teacher model results in the suppression of class correlation information, leading to degradation of the distillation performance. As a result, training a fully converged teacher may not be the optimal choice, especially under resource-limited circumstances. To save training cost, we empirically suggest that the half-way teacher model can suffice. To achieve better distillation, we further proposed an optimal intermediate model selection algorithm to find the appropriate intermediate teacher. Furthermore, this work implies a more economical and efficient way to construct a snapshot ensemble with several intermediate models from the same training trajectory instead of the standard ensemble with independently full-trained models. This technique can significantly improve the ensemble model's distillation performance and reduce the training cost.

Our study also has some limitations. First, the selection of an optimal intermediate model considers the information entropy of the teacher but ignores the variation of the student structures, which can not ensure the optimal KD performance for all teacher-student pairs. Second, how to choose the best intermediate teacher model for a specific structure of student is still a challenging problem.

## Acknowledgement

This work is supported in part by the National Key R&D Program of China under Grant 2020AAA0105200, the National Natural Science Foundation of China under Grants 62022048, THU-Bosch JCML and Beijing Academy of Artificial Intelligence. We also appreciate the generous donation of computing resources by High-Flyer AI.

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
