# A Exploratory experiments

## A.1 Datasets and experimental settings

The "Intermediate Teacher vs. Full Teacher" experiment is conducted on the CIFAR-100 [36] and ImageNet [38] datasets. The CIFAR-100 contains 50,000 training images with 500 images per class and 10,000 test images with 100 images per class, and comprises $32 \times 32$ pixel RGB images with 100 classes. The ImageNet dateset contains 1.2 million images for training and 50,000 for validation from 1,000 classes.

On CIFAR-100, we use a standard data augmentation scheme [44], in which the images are zero-padded with 4 pixels on each side, randomly cropped to produce $32 \times 32$ images, and horizontally mirrored with probability 0.5. WRN-40-2 [39] and ResNet-110 [40] are adopted as teacher models, while WRN-40-1 [39], ResNet-32 [40], and MobileNetV2 [41] are adopted as student models. In this paper, we use MobileNetV2 with its width multiplier set to 0.75. We train each teacher model for 200 epochs with batch size 128, cosine learning rate schedule [45] gradually decaying from 0.1 to 0, weight decay 5e-4, and SGD optimizer with momentum 0.9. We save the intermediate models at the $20^{th}$, $40^{th}$, ..., $180^{th}$ epoch as "Intermediate Teachers", and the final models at the $200^{th}$ epoch as "Full Teachers".

On ImageNet, we adopt MobileNetV2 [41], ResNet-18 [40] as student models and ResNet-50 [40], ResNet-34 [40] as teacher models. We follow the standard PyTorch practice but train teacher models for 120 epochs. The $120^{th}$ checkpoints are taken as the full teachers while the $60^{th}$ checkpoints are adopted as the intermediate teachers.

We distill the student models by the same way as [2] except for the hyperparameters. To perfectly show the performance of each network, we search for the optimal hyperparameters (*i.e.*, the loss ratio $\alpha$ and the temperature $\tau$) to each teacher-student pair as shown in Table 4. Generally, intermediate models at the later training stages tend to choose a larger $\tau$ and a smaller $\alpha$.

The "Snapshot Ensemble vs. Full Ensemble" experiment is conducted on CIFAR-100 [36] and Tiny-ImageNet [37]. Tiny-ImageNet consists of a subset of ImageNet dataset. There are 100,000 images for training and 10,000 images for validation from 200 classes. All images are $64 \times 64$ colored ones. Some different settings on Tiny-ImageNet include: 1) we train models for 150 epochs on Tiny-ImageNet while 200 epochs on CIFAR datasets; 2) we add a stride of 2 to the first layer of the CIFAR models, in order to downsample the images to the same $32 \times 32$ resolution, following [13]. For fair comparison, we also use grid search to find the best value of hyperparameters $\tau$ and $\alpha$, as shown in Table 5.

## A.2 More visual comparison results

Figure 3 only shows the results of total computational cost (teacher and student together) vs. distillation performance between Snapshot Ensemble and Full Ensemble. To show it more comprehensively, we add to show the results of total computational cost vs. distillation performance for five teacher models ($T^{\text{inter}}$, $T^{\text{full}}$, $T^*$, $T_1^{\text{inter}} + T_1^{\text{full}}$, $T_1^{\text{full}} + T_2^{\text{full}}$) on four teacher-student pairs. As shown in Figure 8, the curve of $T^{\text{inter}}$ is on the upper left of that of $T^{\text{full}}$, and the curve of $T_1^{\text{inter}} + T_1^{\text{full}}$ is on the upper left of that of $T_1^{\text{full}} + T_2^{\text{full}}$. In Figure 8, "upper left" means lower computational cost but higher distillation performance. $T^*$ has higher distillation performance than $T_1^{\text{full}}$ and $T_1^{\text{inter}}$, but $T^*$ needs higher computational cost (though lower than $T_1^{\text{full}} + T_2^{\text{full}}$). The current optimal intermediate model selection algorithm needs additional computational cost, which means that there is plenty of room for improvement. A practical suggestion is: in the case of limited computing resources, the half-way teacher model (*i.e.*, $T^{0.5}$) can suffice for KD. In the case of sufficient computing resources, the optimal intermediate model selection algorithm can be used to find an appropriate checkpoint to achieve better performance.

## A.3 Effects of early stopping

In the paper, we training teacher models for a fixed number of epochs, such as 200 epochs on CIFAR, 120 epochs on ImageNet. In practice, the early stopping strategy is often used to avoid model overfitting. To investigate the possible effects of early stopping, we adopt the general early stopping strategy (patience=10) in the training process. We show the training curves of all teacher models

Table 4: Optimal hyperparameters (temperature $\tau$ and ratio $\alpha$) for all used intermediate and full teachers on CIFAR-100 and ImageNet.

| Dataset | T | WRN-40-2 | | ResNet-110 | |
| | S | WRN-40-1 | MobileNetV2 | ResNet-32 | MobileNetV2 |
|---|---|---|---|---|---|
| CIFAR-100 | $T^{20}$ | $\tau$=5, $\alpha$=0.7 | $\tau$=5, $\alpha$=0.7 | $\tau$=5, $\alpha$=0.9 | $\tau$=5, $\alpha$=0.7 |
| | $T^{40}$ | $\tau$=5, $\alpha$=0.7 | $\tau$=5, $\alpha$=0.7 | $\tau$=5, $\alpha$=0.9 | $\tau$=5, $\alpha$=0.7 |
| | $T^{60}$ | $\tau$=5, $\alpha$=0.7 | $\tau$=5, $\alpha$=0.7 | $\tau$=5, $\alpha$=0.9 | $\tau$=7, $\alpha$=0.5 |
| | $T^{80}$ | $\tau$=5, $\alpha$=0.7 | $\tau$=9, $\alpha$=0.5 | $\tau$=5, $\alpha$=0.9 | $\tau$=7, $\alpha$=0.5 |
| | $T^{100}$ | $\tau$=7, $\alpha$=0.7 | $\tau$=9, $\alpha$=0.5 | $\tau$=7, $\alpha$=0.7 | $\tau$=7, $\alpha$=0.5 |
| | $T^{120}$ | $\tau$=7, $\alpha$=0.7 | $\tau$=9, $\alpha$=0.5 | $\tau$=7, $\alpha$=0.7 | $\tau$=7, $\alpha$=0.5 |
| | $T^{140}$ | $\tau$=9, $\alpha$=0.5 | $\tau$=9, $\alpha$=0.5 | $\tau$=7, $\alpha$=0.7 | $\tau$=9, $\alpha$=0.3 |
| | $T^{160}$ | $\tau$=9, $\alpha$=0.5 | $\tau$=9, $\alpha$=0.3 | $\tau$=9, $\alpha$=0.7 | $\tau$=9, $\alpha$=0.3 |
| | $T^{180}$ | $\tau$=9, $\alpha$=0.5 | $\tau$=15, $\alpha$=0.1 | $\tau$=9, $\alpha$=0.5 | $\tau$=9, $\alpha$=0.3 |
| | $T^{200}$ | $\tau$=9, $\alpha$=0.5 | $\tau$=15, $\alpha$=0.1 | $\tau$=9, $\alpha$=0.5 | $\tau$=9, $\alpha$=0.3 |

| Dataset | T - S | ResNet-34 - ResNet-18 | ResNet-50 - MobileNetV2 |
|---|---|---|---|
| ImageNet | $T^{20}$ | $\tau$=1.5, $\alpha$=0.5 | $\tau$=1.5, $\alpha$=0.7 |
| | $T^{40}$ | $\tau$=1.5, $\alpha$=0.5 | $\tau$=1.5, $\alpha$=0.7 |
| | $T^{60}$ | $\tau$=1.5, $\alpha$=0.3 | $\tau$=1.5, $\alpha$=0.5 |
| | $T^{80}$ | $\tau$=2, $\alpha$=0.3 | $\tau$=1.5, $\alpha$=0.5 |
| | $T^{100}$ | $\tau$=2, $\alpha$=0.1 | $\tau$=2, $\alpha$=0.5 |

Table 5: Optimal hyperparameters (temperature $\tau$ and ratio $\alpha$) for Snapshot Ensemble and Full Ensemble on CIFAR-100 and Tiny-ImageNet.

| Dataset | T | WRN-40-2 | | ResNet-110 | |
| | S | WRN-40-1 | MobileNetV2 | ResNet-32 | MobileNetV2 |
|---|---|---|---|---|---|
| CIFAR-100 | Snapshot Ensemble | $\tau$=5, $\alpha$=0.5 | $\tau$=7, $\alpha$=0.3 | $\tau$=9, $\alpha$=0.5 | $\tau$=9, $\alpha$=0.3 |
| | Full Ensemble | $\tau$=5, $\alpha$=0.5 | $\tau$=20, $\alpha$=0.1 | $\tau$=9, $\alpha$=0.5 | $\tau$=9, $\alpha$=0.3 |

| Dataset | T - S | WRN-40-2 - WRN-40-1 | WRN-40-2 - MobileNetV2 |
|---|---|---|---|
| Tiny-ImageNet | Snapshot Ensemble | $\tau$=3, $\alpha$=0.5 | $\tau$=3, $\alpha$=0.5 |
| | Full Ensemble | $\tau$=5, $\alpha$=0.5 | $\tau$=5, $\alpha$=0.5 |

in Figure 9. Overall, training the teacher models on CIFAR for 200 epochs and ImageNet for 120 epochs does not lead to model overfitting. The optimal checkpoints are located on the left of the positions of early stopping. Therefore, using the early stopping strategy does not affect the results.

# B Mutual information experiments

## B.1 Estimation of mutual information

Mutual information is difficult to calculate accurately, especially in the case of unknown joint probability distribution or continuous random variables. In this paper, we estimate the mutual information $I(X; F)$ between input $X$ and representation $F$ with a reconstruction loss following

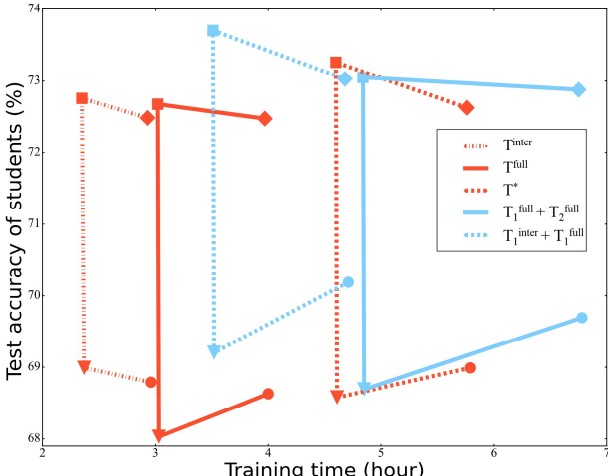

Figure 8: Computational cost vs. distillation performance for different teacher models on CIFAR-100. $T^{\text{inter}}$ denotes a half-way teacher. $T^{\text{full}}$ denotes a convergent teacher. $T_1^{\text{inter}} + T_1^{\text{full}}$ denotes Snapshot Ensemble teacher. $T_1^{\text{full}} + T_2^{\text{full}}$ denotes Full Ensemble teacher. $T^*$ denotes the optimal checkpoint obtained by our search algorithm. □ denotes WRN-40-2/WRN-40-1. △ denotes WRN-40-2/MobileNetV2. ◇ denotes ResNet-110/ResNet-32. ○ denotes ResNet-110/MobileNetV2.

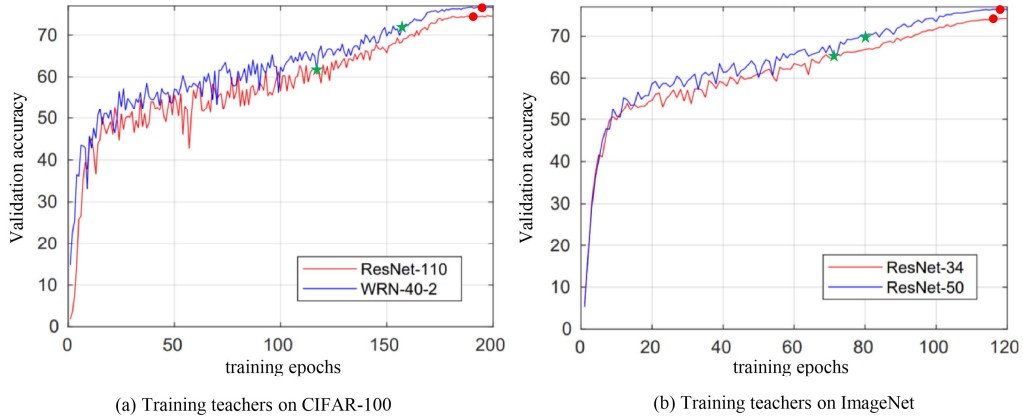

(a) Training teachers on CIFAR-100           (b) Training teachers on ImageNet

Figure 9: The training curves of all teacher models. (a) WRN-40-2 and ResNet-100 trained on CIFAR-100. (b) ResNet-50 and ResNet-34 trained on ImageNet. The red ● denotes the position of early stopping, while the green ⋆ denotes the position of the optimal checkpoints.

[46]. Specifically, we connect a decoder to the last convolution layer of a trained and fixed network model in order to generate a pseudo input image $X$. The structure of the decoder is shown in Table 6). Then we train the decoder to convergence with the Adam optimizer and binary cross-entropy loss between $\overline{X}$ and $X$. This reconstruction loss is used to estimate $I(X; F)$. For the mutual information $I(Y; F)$ between output $Y$ and representation $F$, we use the trained network model to do inference on the test dataset, and estimate $I(Y; F)$ with the test accuracy.

## B.2 More results of class correlation information

The results of heatmaps in Figure 5 show that the appropriate intermediate model has better diversity than the full model. To clearly display differences of class correlation information between the intermediate model and the full model, we randomly sample four classes from the test dataset and calculate the average logits output of each class. Figure 10 shows that the logits output of the

Table 6: Architecture of the decoder.

| Input: $240 \times 240$ / $128 \times 128$ / $64 \times 64$ feature maps |
| :---: |
| Bilinear Interpolation to $32 \times 32$ |
| $3 \times 3$ conv., stride=1, padding=1, output channels=12, BatchNorm+ReLU |
| $3 \times 3$ conv., stride=1, padding=1, output channels=3, Sigmoid |

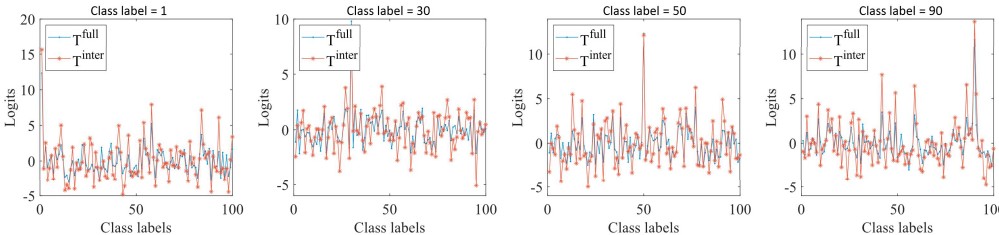

Figure 10: The average logits outputs of 4 random classes on the intermediate model (at the $100^{\text{th}}$ epoch) and the full model (at the $200^{\text{th}}$ epoch). The red curves have more peaks and greater variances than the blue curves, which indicates that $T^{\text{inter}}$ has more non-target class information than $T^{\text{full}}$.

intermediate model has more peaks and larger variance than that of the full model. Specifically, the intermediate model reserves plentiful valuable non-target "misclassifications", which are mostly eliminated in the full model. Such non-target information implicitly illustrates certain correlation among classes thus significantly complements the rigid one-hot label.