# OpenReview forum: "Efficient Knowledge Distillation from Model Checkpoints"
_NeurIPS.cc/2022/Conference — NeurIPS 2022 Accept_

### Official Review · Reviewer_So7a · 2022-06-26

**Rating:** 6
**Confidence:** 2
**Soundness:** 3 good
**Presentation:** 4 excellent
**Contribution:** 2 fair

**Summary:**

The paper deals with knowledge distillation, which is the training of smaller “student” models from larger “teacher” ones in an effort to reduce the model size while preserving as much performance as possible.

In this paper, the authors challenge the notion that better performing (and closer to convergence) models make better teachers, instead showing that checkpoints sampled throughout training make better teachers. This counterintuitive finding is explained using the information bottleneck (IB) principle, which suggests that the mutual information between the parameters of a neural network and the model input increase at the start and then decrease. Informally, this corresponds to the observation that models begin by learning correlates at the start, then learn to discard information about class correlates that would be helpful for knowledge distillation. The authors show that teachers are improved by constructing them from “intermediate” models (i.e. checkpointed versions of a model sampled as it is trained) instead of using converged models.

Further, they use the IB principle to identify the optimal intermediate model for knowledge distillation. They show that selecting a model that maximizes the sum of the mutual information between the parameters and input and the mutual information between the parameters and output yields better intermediate teachers than simpler schemes that sample models at fixed training intervals.


**Questions:**

How significant are the results? (Equivalently, if you were to randomly initialize your experiments and run them again and again, how often would $T^*$ be better than the alternatives.)

**Limitations:**

Improvement is shown in percentage points, but there is no information presented on the standard deviation of these results. This means that we cannot conclude that the change in results is statistically significant. To make this concrete: in Table 3, $T^*$ has an average performance of 70.87%, which is presented as better than $T^{0.7}$ at 70.77%. It is not clear to me that this difference of 0.1% is better, particularly on the CIFAR-100 dataset.

The way to fix this is to run each experiment multiple times, present the result and standard deviation, and bold the results that are within some confidence interval of the top result. I suspect that repeating the analysis with this addition will show that the results are less significant than claimed.

Unfortunately, the experiments showing that the use of snapshot teachers/ensembles is better than full, and the experiments establishing the usefulness of their proposed algorithm are also vulnerable to this criticism. I believe this is a fatal flaw in the paper, and needs remediation before publication.

(Similar changes should also be incorporated into the construction of Figure 4. It is not clear to me how significant the shift in the blue vs gray lines are unless some measure of deviation can be plotted in the chart.)

Consider running these experiments again to establish that they are significant. If you cannot show that, consider investigating how a smaller training budget for the teacher network is enough to give good performance in student networks. That may be a useful result.

**Strengths And Weaknesses:**

# Originality
The work is reasonably original. It is the first application of the IB principle to Knowledge Distillation (KD), yielding an observation that improves the state of KD.

# Quality
The interpretation of IB presented in the paper is compelling, but the experiments don’t provide enough information to confirm the significance of the results. I explain this in the Limitations section of the review.

# Clarity
The writing is clear and proceeds logically from section to section. I particularly like the intuitive introduction to IB (lines 44-63), which clearly gives readers sufficient intuition to understand the work. I also really appreciate how the related work section summarizes the rather extensive KD literature as a coherent narrative.

Minor points:
 - In Figure 2, the yellow line (ResNet-110/MobileNetV2) is unreadable. Consider sampling colors of different hues with a fixed saturation and value/lightness to prevent such issues.
 - Avoid floating figures on one side of the page, use the default setting of a full-width figure instead. This greatly improves the flow of text.
Footnote 2 (link to tiny-imagenet) can be replaced by a citation – bibtex supports URL citations. This should also save you a little vertical space for the publication-ready version of the paper.
 - Line 152 “Fistly” should read “Firstly”

# Significance

The results are sufficiently significant, as it is likely to spawn a line of follow-up work expanding our understanding of KD using IB principles. This is conditional on the results being statistically significant, though.

---

> ### Author Response · Authors · 2022-07-28
> **Responses to Reviewer So7a**
>
> Thanks for your valuable comments. We carefully respond to the weaknesses you have pointed out.
>
> > how significant are the results?
>
> We apologize for the lack of standard deviations. In fact, we had calculated the standard deviations, but we removed them from the manuscript in consideration of the fact that the limited width of the table causes the data font to be too small. Due to limited space, we have supplemented the whole standard deviation data in the rebuttal revision pdf. We show some typical data in Table 1 and Table 2. **Overall, the standard deviation data does not obscure the advantage of the intermediate model over the convergent model. Even if some of the improvements are not significant, the significant decrease in training cost is still valuable.**
>
> Table 1. Evaluation of distillation performance of all checkpoints on CIFAR. Results within 95% confidence interval of the best results are in bold.
> |Teacher|Student|$T^{20}$|$T^{40}$|$T^{60}$|$T^{80}$|$T^{100}$|$T^{120}$|$T^{140}$|$T^{160}$|$T^{180}$|$T^{200}$|
> |:---------|:---------|:---------|:--------|:---------|:---------|:---------|:---------|:---------|:---------|:---------|:---------|
> |WRN-40-2|WRN-40-1|71.65±0.28|72.21±0.13|72.34±0.1|72.85±0.15|72.76±0.24|72.83±0.06|73.08±0.05|**73.26±0.03**|72.91±0.27|72.68±0.1|
> |ResNet-110|ResNet-32|70.68±0.49|70.74±0.18|70.98±0.07|71.44±0.13|72.49±0.32|**72.63±0.13**|**72.56±0.30**|72.49±0.28|**72.53±0.13**|72.48±0.22|
> |WRN-40-2|MobileNetV2|67.77±0.26|67.86±0.22|68.21±0.33|**68.94±0.3**|**68.99±0.12**|68.74±0.21|68.54±0.07| 68.58±0.34|68.19±0.35| 68.03±0.34|
> |ResNet-110|MobileNetV2|66.38±0.29|67.88±0.17|67.84±0.26|68.66±0.07|68.79±0.17|**68.99±0.33**|**69.01±0.20**|**69.05±0.27**|**68.84±0.52**|68.63±0.35|
>
> Table 2. Comparison results of ''Full Ensemble vs. Snapshot Ensemble'' on CIFAR. Distillation results within 95% confidence interval of the best results are in bold. The best ensemble results are italic.
>
> |Teacher|Student|Normal KD|Full Ensemble KD|Snapshot Ensemble KD|Full Ensemble|Snapshot Ensemble|
> |:---------|:---------|:---------|:--------|:---------|:---------|:---------|
> |WRN-40-2|WRN-40-1|71.68±0.10|73.05±0.16|**73.70±0.04**|*79.44*|76.29|
> |ResNet-110|ResNet-32|72.48±0.22|**72.88±0.14**|**73.03±0.24**|*76.92*|73.23|
> |WRN-40-2|MobileNetV2|68.03±0.34|68.69±0.32|**69.20±0.31**|*79.44*|76.29|
> |ResNet-110|MobileNetV2|68.63±0.35|69.69±0.25|**70.19±0.25**|*76.92*|73.23|
>
> In Table 3 of the paper, $T^*$  has an average performance of 70.87%, which is better than $T^{0.7}$ at 70.77%. The improvement looks minor because $T^{0.7}$ is a strong baseline. As stated in our paper, the optimal checkpoints are usually found in the upper right corner of the IB curve. We think it is more valuable that both $T^{*}$ and $T^{0.7}$ are clearly better than $T^{full}$ (70.46%).
>
> As you said, "consider investigating how a smaller training budget for the teacher network is enough to give good performance in student networks. That may be a useful result." We agree with it and have claimed that in the conclusion section. **In the case of limited computing resources, we empirically suggested that the half-way teacher model (i.e., $T^{0.5}$) can suffice for KD. In the case of sufficient computing resources, we further proposed an optimal intermediate model selection algorithm to find an appropriate checkpoint to achieve better performance.**
>
> > Some minor points on clarity.
>
> We have polished the paper, including changing the color of the lines in Figure 2, replacing footnotes with citations, and eliminating spelling errors. We use some floating figures to ensure the close connections between texts and images so that readers easily read them.

---

> > ### Comment · Reviewer_So7a · 2022-08-03
> > **Appreciate the extended results, and a note on significance.**
> >
> > I appreciate the extended results table, and I have increased my soundness rating accordingly. (For clarity, you should bold all results where the mean lies within the, for example, 5% lower-confidence-bound of the top result. This corresponds to results that are not less than 5% likely to be better than your top result, and is the correct way to ensure that model comparisons are not spurious.)
> >
> > Now that the standard deviations have been added to the paper, I am more interested in the significance of the contribution. There are two ways (that I can think of) to establish this; if you can think of a different way feel free to pursue that:
> >
> > 1. Show a noticeable improvement in the limited compute case, perhaps by plotting the total amount of compute (for teacher and student training together) vs the final model performance and showing how your method, with the optimal intermediate model selection algorithm, achieves similar results in less time than full ensemble KD. This uses data already in your tables, so you won't need to run any additional experiments, but presenting it in a coherent way will make your case much more convincing.
> >
> > 2. Show a noticeable improvement in performance over the baselines identified by reviewer encu. This would show that your method is better than the many existing variations of KD.
> >
> > (Also, after reading the other reviewer's responses, I have decreased my own confidence score.)

---

> > > ### Author Response · Authors · 2022-08-04
> > > **Responses to the note on significance**
> > >
> > > Thanks for raising your score. It encourages us a lot.
> > >
> > > >How to represent the comparision results.
> > >
> > > Following your advice, we bold the results within 0.95 confidence interval of the top results, as shown in Table 1 and Table 2 above. Generally, the bold results come from intermediate checkpoints rather than convergent models, and Snapshot Ensemble has better distillation performance than Full ensemble.
> > >
> > > >How to show the significance of the contribution.
> > >
> > > (1) It's a good suggestion to show the comprehensive comparisons of different teacher models by plotting the total computational cost (teacher and student together) vs the distillation performance. Figure 3 in the paper only showed this time-performance comparisons between the Snapshot Ensemble and the Full Ensemble. To show it more comprehensively , we further illustrate the results of computational cost vs distillation performance for five teacher models ($T^{inter}$, $T^{full}$, $T^*$, $T_1^{inter}+T_1^{full}$, $T_1^{full}+T_2^{full}$) on four teacher-student pairs. These results have been provided in the supplementary materials (see Section 4 and Figure 4). As shown in Appendix Figure 4, the curve of $T^{inter}$ is to the upper left of the curve of $T^{full}$, and the curve of $T_1^{inter}+T_1^{full}$ is to the upper left of the curve of $T_1^{full}+T_2^{full}$. In the figure, "upper left" means lower computational cost but higher distillation performance. $T^*$ has higher distillation performance than $T^{full}$ and $T^{inter}$, but $T^*$ needs larger computational cost (though lower than $T_1^{full}+T_2^{full}$). As we said in "Shared responses", the algrithm of $T^*$ needs to be further improved in the future work.
> > >
> > > (2) We agree that our method can be better than many existing variations of KD. In fact, "Contrastive representation distillation [ICLR 2020]" [1] evaluated many algorithms and found that "KD works pretty well and none of the other methods consistently outperforms KD." (see Table 1 and Table 2 of [1]) Many works adopted the best hyperparameters to highlight their own methods, which is unfair to normal KD. However, we did the optimal hyperparameter search for every teacher-student pair in this paper. Our work is only based on normal KD because we try to obtain a common and useful conclusion and avoid any unnecessary trick.
> > >
> > > ---
> > > [1] Tian, Yonglong, Dilip Krishnan, and Phillip Isola. "Contrastive Representation Distillation." International Conference on Learning Representations. 2020.

---

### Official Review · Reviewer_6Nks · 2022-07-05

**Rating:** 6
**Confidence:** 4
**Soundness:** 3 good
**Presentation:** 3 good
**Contribution:** 3 good

**Summary:**

The authors propose a new model distillation method utilizing teacher's training checkpoints, showing performance gain over a handful of benchmarks. The authors connect their methods with information bottleneck theory and hypothesize that the performance gain partly comes from the fact that checkpoints save a lot of representation information about inputs, without overfitting to the task. Although the findings are somewhat interesting, I have some concerns with the core methods I discuss below.

**Questions:**

- **Standard deviation is really useful for gaining real insights about performance improvements**
The authors show that multiple experiments over 5 random seeds are conducted. Can you include the standard deviation in all main results tables? Since the performance increment looks small numerically, grounding them with SD could be very useful in interpreting the results.

- **Instead of showing the T^{0.3} T^{0.5} and T^{0.7}, would it be possible to evaluate all checkpoints as in Fig. 4**
It would be still interesting to observe that IF curves completely correlate with the downstream distilled model performance.

- **Worth exploring checkpoint distillation with language models**
If time permitted, I would be interested to see if this works with language models as well.

**Ethics Review Area:**

["I don’t know"]

**Limitations:**

The authors adequately addressed the limitations and potential negative societal impact.

**Strengths And Weaknesses:**

**Strength**:
- The connections between the information bottleneck theory and model distillation seem intriguing. However, I am not convinced that this is the real cause of the performance gain.
-  It is interesting to see how checkpoints (i.e., not overfitting the training data) generalize better in model distillation.

**Weakness**:
- **The fact that the information bottleneck theory shows correlated behavior with distilled model performance does not mean IF is the cause or the explanation behind the result**

The core contribution of the paper is providing a checkpoint selection method based on IF for the teacher model. The core claim behind this contribution is that IF offers a way to measure how good a checkpoint is in preserving "dark information" which leads to better image representations for model distillation. However, it is not convincing to me that there is a causal link between IF and checkpoint distillation. Fig.4 is somewhat trivial to any pretrain-then-finetune paradigm, i.e., in the initial phase of finetuning, the pretrained model begins with poor in task performance but with a good representation to reconstruct outputs; at the end, the model becomes overfitting to the task with sacrificed reconstruction accuracies. The fact that IF seems to correlate with downstream distilled model performance does not necessarily prove IF is the reason behind this phenomenon. I layout another very simple theory which could be another explanation behind this, which is in fact, proved in the literature as well.


- **What if it is just the fact that the pertaining objective helps in model distillation? i.e., earlier checkpoints simply provide closer weights to the pretrained model, which provides more information about the pertaining objective**

Using the checkpoints to distill can simply be viewed as a way of preventing the student to overfit the downstream finetuning task. Specifically, it asks the student model to essentially forget task-specific information and asks the student model to recover pretrained information. This is indeed proved by some recent works in language model distillation, i.e., while distilling with the task objective, the student model is also co-trained with the teacher's pretraining objective. With this, there could be many other baselines to run, what if we simply distill with lower learning rates, or with larger learning rate decays? What if we do weighted distillation loss by combining task-specific loss as well as pertaining loss?

---

> ### Author Response · Authors · 2022-07-28
> **Responses to Reviewer 6Nks**
>
> Thanks for your valuable comments. We carefully respond to the weaknesses and  questions.
>
> > W1: what if it is just the fact that the pretraining objective helps in model distillation?
>
> Some mentioned terms ("pretraining objective", "closer weights", "pretrain-then-finetune", etc.) are not so suitable to our task setting, because the settings of several tasks: classical KD, language model pretrain-finetune and language model distillation are quite different. We try to use mathematical language to explain the differences. Firstly, we define $M_{t}$ as the teacher model, $M_s$ as the student model, $T_I$ as the image classification task, $T_{LP}$ as the language model pretraining task, $T_{LF}$ as the language model fine-tune (downstream) task.
>
> (1) Classical KD (proposed by Hinton [13], our setting). It follows a "pretrian-distillation" paradigm. Firstly, a randomly initialized teacher model $M^0_t$ is trained on $T_I$ with the cross entropy loss $L_{ce}$ to obain $M^1_t$. Secondly, a randomly initialized student model $M^0_s$ is also trained on $T_I$ with $L_{ce}$ and the distillation loss $L_{dis}(M^1_t)$ to obain $M^1_s$. This task can be formalizad as: $M^0_t\xrightarrow[T_I]{L_{ce}} M^1_t, M^0_s\xrightarrow[T_I]{L_{ce}, L_{dis}(M^1_t)} M^1_s$. (1)
>
> (2) Language model pretrain-finetune task. Firstly, a randomly initialized model $M^0_t$ is trained on $T_{LP}$ with a self-supervised pretrain objective $L_{pre}$ to obtain $M^1_t$. Secondly, $M^1_t$ is fine-tuned on $T_{LF}$ with the downstream task-sepcific objective $L_{down}$ to obtain $M^2_t$. There is no student model in the task. This task can be formalizad as:
> $M^0_t\xrightarrow[T_{LP}]{L_{pre}} M^1_t, M^1_t\xrightarrow[T_{LF}]{L_{down}} M^2_t$. (2)
>
> (3) Language model distillation task (such as DistillBert, TinyBert). There has been a pretrained large teacher $M^1_t$ (a full Bert), a lightweight student $M^1_s$ with fewer layers of the same weights. In the pretrain distillation stage, $M^1_s$ is distilled by $M^1_t$ on $T_{LP}$ to obtain $M^2_s$. In the fine-tune distillation stage, $M^1_t$ is fine-tuned on $T_{LF}$ to obtain $M^2_t$, and then $M^2_s$ is ditilled by $M^2_t$ on $T_{LF}$ to obtain $M^3_t$. This task can be formalizad as:
> $M^1_s\xrightarrow[T_{LP}]{L_{pre},L_{dis}(M^1_t)} M^2_s, M^2_s\xrightarrow[T_{LF}]{L_{down},L_{dis}(M^2_t)} M^3_s$. (3)
>
> **Comparing the formulas (1)(2)(3), the three task settings are clearly different.** Our setting does not follow the "pretrain-then-finetune" paradigm. The "pretraining objective" in our task is the cross-entropy loss which obviously doesn't help distillation.
>
> > W2: the fact that IB theory shows correlated behavior with distilled model performance does not mean IB is the cause or the explanation behind the result.
>
> **In our view, IB theory is tailored to explain KD, and it is not farfetched to link IB with KD.** Looking at formula (1), $L_{ce}$ contains complete information of the category label Y, which means sufficient $I(F; Y)$. $L_{dis}(M^1_t)$ helps to improve students' performance because it contains information about input X, i.e., $I(X; F)$. Therefore, It is logical that using $I(X; F)$ to explain "dark knowledge". A good distillation means a good balance between $I(X; F)$ and $I(F; Y)$. Therefore, It is reasonable to connect IB with KD and search better teacher checkpoints by IB cures. Figure 4 are not trivial and obvious because our task setting does not follow "pretrian-then-finetune" paradigm.
>
> > Q1: evaluate all checkpoints.
>
> In fact, the distillation performance of all checkpoints had been shown in Figure 2. We also show it in table style (Due to limited space, please see Table 2 in "Responses to Reviewer encu"). **The distillation performance of teacher checkpoints generally increases first and then decreases. This trend is consistent with the trend of I(X; F) in the IB curve, which is one of the intuitive reasons why we connect IB theory with KD.**
>
> > Q2: lack of the standard deviation.
>
> It is a good suggestion to show the standard deviation data. Due to limited space, we have supplemented the whole standard deviation data in the rebuttal revision pdf. You can see some typical data in Table 2 of "Responses to Reviewer encu". **Overall, the standard deviation data does not obscure the advantage of the intermediate model over the convergent model. Even if some of the improvements are not significant, the significant decrease in training cost is still valuable.**
>
> > Q3: worth exploring checkpoint distillation with language models.
>
> Almost all of the papers included in our related work did not investigate language models. We agree that it is tempting to apply our approach to large-scale language models. However, exploring language models is too costly for us in terms of computational resources and training time, because we need to train large-scale language models from scratch, such as Bert. It is too difficult to investigate the results in rebuttal time.

---

> > ### Comment · Reviewer_6Nks · 2022-08-08
> > **Thank you for the clarifications. I'm raising my score.**
> >
> > Thank you for the clear answers and the added experiments. The responses to my comments as well as others resolve my question about the differences between this work and other model distillation in the NLP regime. Thus, I am raising my score significantly, from 4 to 6.

---

> ### Author Response · Authors · 2022-08-06
> **Have our responses addressed your concerns?**
>
> Dear Reviewer 6Nks:
>
> Thank you for raising the weaknesses and questions. We have tried to address them in our responses below and rebuttal revision pdf. We would like to know if our responses address your concerns. If not, we would be happy to provide more explanation. Additional suggestions or discussions are also welcome.
>
> Best wishes.

---

### Official Review · Reviewer_encu · 2022-07-10

**Rating:** 6
**Confidence:** 4
**Soundness:** 3 good
**Presentation:** 3 good
**Contribution:** 3 good

**Summary:**

This paper observes that the intermediate models (checkpoints in the middle of the training procedure) can serve as better teachers than fully trained teacher models in knowledge distillation (KD).
Meanwhile, this paper proposes the Snapshot Ensemble, which ensembles several intermediate models and one full model, and observes that the snapshot ensemble has better KD performance and low training cost than the full teacher ensemble.
Furthermore, this paper applies the Information Bottleneck (IB) theory to understand KD and “dark knowledge”. Moreover, this paper proposes an algorithm to find the optimal intermediate teacher based on IB theory.


**Questions:**

The idea of applying IB theory to KD is in general novel. Nevertheless, the above weaknesses refrain me from giving a better score. I would be happy to improve my rating as long as the authors address the above weaknesses in the rebuttal.


**Limitations:**

Yes, this paper does mention some limitations in Section 6.

**Strengths And Weaknesses:**

Strengths.
- The idea of applying mutual information to understand KD and “dark knowledge” is very novel to the community.
The curves of mutual information between the input and learned features clearly reveal the training process of a neural network and dark knowledge.

- Experiments are sufficient.

- This paper is in general well written and easy to follow.

Weaknesses

- The findings of the intermediate checkpoints are not very exciting to the community.
My most concern is about the novelty of the intermediate model part. This paper spends a lot of effort demonstrating the advantages of the intermediate checkpoints. However, previous works [36] already suggest that the Reversed KD (“student teach the teacher”) or even Defective KD (“poorly-trained teacher teach the student”) can achieve comparable results with normal KD. Thus, the observations on intermediate models in this paper should be naturally correct as well and not very exciting. Although this paper claims that the training cost of a teacher is significantly lower, the defective teacher [36] can further reduce training effort. To this end, I would suggest this paper compare their methods with methods in [36], such as defective-KD, reversed-KD, and teacher-free KD. Specifically, this paper can compare the accuracy and total training cost with [36] on the experiments in Tables 1, 2, and 3. With these additional experiments, this paper can convincingly claim the advantages of the intermediate checkpoints. Otherwise, I would suggest this paper turns the tune down on the intermediate models and put more effort into the IB theory section.

- The error bar is missing.
Although this paper claims that the results is evaluated in five independent experiments in CheckList 3(c), this paper only reports the average value, while the standard deviation (error bar) is missing. Since the improvement in many experiments is relatively small (less than 0.5), the experimental results can not be significant until this paper demonstrate that the improvement is large than the error bar.

---

> ### Author Response · Authors · 2022-07-27
> **Responses to Reviewer encu**
>
> Thanks for your positive comments. We carefully respond to the weaknesses.
>
> > W1: the differences between "Revisiting Knowledge Distillation via Label Smoothing Regularization" [36] and our work. The novelty of the intermediate model part.
>
> The previous work [36] is one of the most related works to our research. We had done in-depth researches and analyses on this article, and obtained some conclusions.
>
> (1) We agree that the exploring of Defective KD which uses an early teacher checkpoint to distill the student is similar to our first exploring experiment. **However, they only claimed that Defective KD could improve students' performance, but could not achieve or exceed normal KD.** Table 1 deriving from [36] (see table 1 and 4 of [36]) supports our viewpoint. Furthermore, [36] adopted almost the same temperature setting (temperature 20, see table 10 and 11 of [36]), which was detrimental to show the performance of baselines. In contrast, we investigated the distillation performance of all teacher checkpoints by searching the optimal hyperparameters (see section 1 of our supplementary material), and found that some specific intermediate models performed better than the convergent model, which was not discovered by [36]. The reversed-KD and teacher-free KD are orthogonal to our exploring.
>
> Table1. Partial results of De-KD (Defective KD) from [36]
>
> |    Teacher   |Student                         |Normal KD |De-KD|
> |----------------|-------------------------------|-----------------------------|-----------------------------|
> |ResNet18: 75.87|MobileNetV2: 68.38          |71.05±0.16        |70.65±0.35|
> |ResNet18: 75.87         |ShufﬂﬂeNetV2: 70.34           |72.05±0.13          |71.82±0.11|
> |ResNeXt29: 81.03          |MobileNetV2: 68.38|71.65±0.41|71.52±0.27|
> |ResNeXt29: 81.03         |ResNet18: 75.87          |77.84±0.15          |77.28±0.17|
>
> (2) In fact, **we had evaluated the performance of De-KD in our paper.** For example, the distillation performance of all checkpoints has been shown in Figure 2, where the distillation using earlier teacher checkpoints (fewer than 60 epochs) can be considered as De-KD. Obviously, De-KD can not achieve the performance of normal KD. We also list the distillation performance of all checkpoints in Table 2 (after optimal hyperparameter search). **Obviously, the best checkpoints do not appear in the early stage. It means De-KD < Normal KD < Ours.**
>
> Table 2. Evaluation of distillation performance of all checkpoints. The best results are bold.
>
> |Teacher|Student: acc|$T^{20}$|$T^{40}$|$T^{60}$|$T^{80}$|$T^{100}$|$T^{120}$|$T^{140}$|$T^{160}$|$T^{180}$|$T^{200}$|
> |:---------|:---------|:---------|:--------|:---------|:---------|:---------|:---------|:---------|:---------|:---------|:---------|
> |WRN-40-2|WRN-40-1:70.38|71.65±0.28|72.21±0.13|72.34±0.1|72.85±0.15|72.76±0.24|72.83±0.06|73.08±0.05|**73.26±0.03**|72.91±0.27|72.68±0.1|
> |ResNet-110|ResNet-32:70.16|70.68±0.49|70.74±0.18|70.98±0.07|71.44±0.13|72.49±0.32|**72.63±0.13**|72.56±0.30|72.49±0.28|72.53±0.13|72.48±0.22|
> |WRN-40-2|MobileNetV2:64.49|67.77±0.26|67.86±0.22|68.21±0.33|68.94±0.3|**68.99±0.12**|68.74±0.21|68.54±0.07| 68.58±0.34|68.19±0.35| 68.03±0.34|
> |ResNet-110|MobileNetV2:64.49|66.38±0.29|67.88±0.17|67.84±0.26|68.66±0.07|68.79±0.17|68.99±0.33|69.01±0.20|**69.05±0.27**|68.84±0.52|68.63±0.35|
>
> (3)  The main contribution of [36] is establishing the connection between KD and LSR (Label Smoothing Regularization), and propose that KD is equivalent to LSR when the temperature value is large. Different from their contribution, **we find that specific intermediate models are more effective than the convergent model from the perspective of retaining the category correlation information, further use IB to explain this phenomenon and establish the connection between IB and KD.**
>
>
> **Therefore, [36] did not fully discover the importance of intermediate models in KD.** On the contrary, we also explained the connection between LSR and KD by using IB theory (see section 2.3 of our supplementary material), which further supported and complemented [36]. We accept the good suggestion of "put more effort into the IB theory section", consider to reorganize and expand this part.
>
> > W2: The error bar is missing.
>
> We apologize for the lack of standard deviations on CIFAR dataset. In fact, we have calculated these data, but we removed them from the manuscript in consideration of the fact that the limited width of the table causes the data font to be too small. Due to limited space, we have supplemented the whole standard deviation data in the rebuttal revision pdf. Partial data has been shown in the Table 2 above. **Overall, the standard deviation data does not obscure the advantage of the intermediate model over the convergent model, while the decrease in training cost is quite clear and valuable.**

---

> ### Author Response · Authors · 2022-08-06
> **Have our responses addressed your concerns？**
>
> Dear Reviewer encu:
>
> Thanks for raising the concerns and weaknesses. We have tried to address them in our responses and rebuttal revision pdf. We would like to know if our responses address your concerns. If not, we would be happy to provide more explanation. Additional suggestions or discussions are also welcome.
>
> Best wishes.

---

> > ### Comment · Reviewer_encu · 2022-08-08
> > **Thanks for your reply**
> >
> > The response clearly solves my concerns about the difference between this work and teacher-free KD. Thus, I improve my final rating from 5 to 6.

---

### Official Review · Reviewer_CyDK · 2022-07-19

**Rating:** 7
**Confidence:** 4
**Soundness:** 3 good
**Presentation:** 2 fair
**Contribution:** 3 good

**Summary:**

**A brief summary**: The paper makes an interesting observation that an intermediate checkpoint of the (teacher) model is just as good for the purposes of knowledge distillation (i.e., training a student model). The paper provides ample evidence to support their observations. They explain this observation using information theory: intermediate checkpoints retain more mutual information (I(X, F)) between the input and the representations than the converged models, which have compressed I(X, F) to maximize target-related information (I(Y, F)). Put simply, intermediate models retain more information about the inputs and other non-target classes. The paper further provides a simple scheme to identify intermediate checkpoints most suited for distillation and demonstrate the efficacy of checkpoints identified through their protocol.

EDIT: Updated my score from 6 $\rightarrow$ 7, based on the responses during the discussion phase.

**Questions:**

How would the results in Table 1 and Table 2 look like if T_{full} is a model based on early stopping rather rather than a model after 120 or 200 epochs?

Please also see the main review for other suggestions on presentation and optimal model-selection algorithm.

**Limitations:**

(Copy pasting my concerns from my main review).

Some of my minor concerns—largely concerning the presentation of the paper—are as follows:

The figure and table captions in the paper are often incomplete and sow confusion. For instance, after reading Figure 2 the impression a reader gets is that we are looking at teacher performance over epochs. Only later in section 4, it becomes clear that the y-axis is the distillation performance of student models for various intermediate teacher models trained for that many epochs (in x-axis). Similarly, Figure 1 is a cartoon representation of the key result and can be misleading (the KD performance is not necessarily as high as figure 1 would have you believe. Further, it is never clarified if the figure stems from real values or is just a high-level cartoon depiction). Similarly, Figures 4 and 5 leave out important details.

The paper title, albeit catchy, doesn’t really concern the paper’s key observations, methods or results and is rather a snappy message which could apply to several papers.

This is more of a suggestion than a concern: it would be great to have an intermediate model-selection process that doesn’t require training the teacher model to convergence (which means that we don’t save any training cost). This could be future work, and therefore could be discussed in the conclusions and limitations section.

Some typos that I noticed and need fixes:

- Line 73, Rethinking → rethinking
- Line 171, comparable or even better (the word or is missing)
- Line 329, s in missing from the limitations
- Line 345, We → we




**Strengths And Weaknesses:**

**My assessment**: Overall, I enjoyed reading the paper and think that the paper would be of interest to other members in the research community. Their results also suggest that one could significantly reduce training cost of the teacher models. The key result is intriguing: an intermediate teacher model—checkpoint corresponding to 50% of full convergence—is less accurate by about 8-14 points, **yet yields comparable distillation performance**. This result is supported through a number of experiments on various student-teacher pairs. Their protocol to select the optimal checkpoint is also simple, reasonable and effective.

Despite my overall positive assessment, I have one major and a few minor concerns. My biggest concern is that selecting 200 or 120 epochs upfront is unsettling. In practice, most training runs use an early stopping criterion, where models are trained to a point till they improve validation performance. I would be more confident in the results and my recommendation if the T_{full} is a model based on early stopping rather rather than a model after 120 or 200 epochs. I am curious to see how the key results presented in Table 1 and 2 change upon making this change?

Some of my minor concerns—largely concerning the presentation of the paper—are as follows:

The figure and table captions in the paper are often incomplete and sow confusion. For instance, after reading Figure 2 the impression a reader gets is that we are looking at teacher performance over epochs. Only later in section 4, it becomes clear that the y-axis is the distillation performance of student models for various intermediate teacher models trained for that many epochs (in x-axis). Similarly, Figure 1 is a cartoon representation of the key result and can be misleading (the KD performance is not necessarily as high as figure 1 would have you believe. Further, it is never clarified if the figure stems from real values or is just a high-level cartoon depiction). Similarly, Figures 4 and 5 leave out important details.


The paper title, albeit catchy, doesn’t really concern the paper’s key observations, methods or results and is rather a snappy message which could apply to several papers.


This is more of a suggestion than a concern: it would be great to have an intermediate model-selection process that doesn’t require training the teacher model to convergence (which means that we don’t save any training cost).  This could be future work, and therefore could be discussed in the conclusions and limitations section.


Some typos that I noticed and need fixes:


- Line 73, Rethinking → rethinking
- Line 171, comparable or even better (the word or is missing)
- Line 329, s in missing from the limitations
- Line 345, We → we

---

> ### Author Response · Authors · 2022-08-01
> **Responses to Reviewer CyDK**
>
> Thanks for your positive comments. We carefully respond to the questions.
>
> > Q1: how would the results in Table 1 and Table 2 look like if $T^{full}$ is a model based on early stopping rather rather than a model after 120 or 200 epochs?
>
> Overall, training the teacher models on CIFAR for 200 epochs and ImageNet for 120 epochs does not lead to obvious overfitting. The numbers of training epochs are not significantly affected by using early stopping. We have tested the common early stopping strategy (patience=10) on all teacher models. Table 1 shows the numbers of training epochs for teacher models with or without early stopping. We also show the curves of validation accuracy versus epoch for all teacher models in the supplementary material (see section 3, Figure 3).  **It shows that whether we use early stopping has no effect on the results of our paper.**
>
> Table 1. The numbers of training epochs for teacher models with or without early stopping.
>
> |Teacher model|$T^{full}$ without early stopping|$T^{full}$ with early stopping|the optimal $T^{inter}$|
> |:---------:|:---------:|:---------:|:---------:|
> |WRN-40-2|200|197|160|
> |ResNet-110|200|189|120|
> |ResNet-50|120|115|80|
> |ResNet-34|120|119|70|
>
> > Q2: the figure and table captions in the paper are often incomplete and sow confusion.
>
> These detailed suggestions are very useful in polishing our paper. We have fixed Figures 1, 2, and 3, such as amending the X-axis and Y-axis captions (see the rebuttal revision pdf). We will keep improving the presentation of figures and tables till submitting the final version.
>
> > Q3: the paper title, albeit catchy, doesn’t really concern the paper’s key observations, methods or results and is rather a snappy message which could apply to several papers.
>
> Thanks, we will reconsider the title of the paper.
>
> > Q4: it would be great to have an intermediate model-selection process that doesn’t require training the teacher model to convergence.
>
> This is a good suggestion. In the case of limited computing resources, we empirically suggested that the half-way teacher model can suffice for KD. In the case of sufficient computing resources, we proposed the optimal intermediate model selection algorithm to find an appropriate checkpoint to achieve better performance. The current algorithm indeed requires training the teacher model to convergence, which does not save training cost. In the next, we will improve the algorithm and also hope that the followers can propose wiser algorithms.
>
> > Q5: some typos need fixes.
>
> We have fixed them (see the rebuttal vision pdf).

---

> > ### Comment · Reviewer_CyDK · 2022-08-08
> > **Thanks for your responses**
> >
> > I would like to thank the authors for their detailed responses and updating the draft in response to some of the suggestions. I enjoyed reading through other reviewers' comments and suggestions, and the authors' responses. Going over the discussion, I had a follow up suggestion and a small clarification. First, given results from [36], I would suggest the authors discuss more prominently their contributions related to linking the KD observation and IB theory (contribution #2 as per the general response from the authors), rather than emphasizing so much on the observation that a fully-converged teacher is suboptimal (which is already known). Second, I would like to clarify how would a practitioner know about $T_{0.5}$  (a half-point) checkpoint without having trained the model (with early stopping)? That is, how would one know if it may take 117 epochs or just 93 to converge, without actually training it.
> >
> > I particularly appreciate the additional experiments to include the error bars, and validate the original hypothesis with early stopping.  The experiments with respect to early stopping alleviate my concerns. (Additionally, Reviewer 6Nks's concern and alternative hypothesis likely doesn't apply here as the authors' don't use pretrained teacher/student models). In that light, I am happy to update my score and recommend this paper for an acceptance.

---

> > > ### Author Response · Authors · 2022-08-09
> > > **Supplementary responses**
> > >
> > > Thanks very much for your continued support.   It encourages us a lot.
> > >
> > > (1) We sincerely accept the constructive suggestion of discussing contribution #2 more prominently.   Some related study contents were put in the supplementary materials (such as, Section 2 and Section 4). We are considering to supplement and reorganize this part in the final version.
> > >
> > > (2) For commonly used models, we suggest that researchers can easily find the proper settings including the training epochs for convergent models, from the published papers or a number of open source sites, such as Github.
> > >
> > > We thank the reviewers again for all their help so far in improving the paper!

---

> ### Author Response · Authors · 2022-08-06
> **Have our responses addressed your concerns？**
>
>
> Dear Reviewer CyDK:
>
> Thank you for raising the concerns and questions.  We have tried to address them in our responses below and rebuttal revision pdf.  We would like to know if our responses address your concerns. If not, we would be happy to provide more explanation.  Additional suggestions or discussions are also welcome.
>
> Best wishes.

---

### Author Response · Authors · 2022-07-27
**Shared responses to all ACs and reviewers**

# Shared responses to all ACs and reviewers

We thank all reviewers for their careful review and valuable comments which help us improve our work. We are delighted to see that **all four reviewers approve our novelty**. This encourages us a lot.

R1: "I enjoyed reading the paper and think that the paper would be of interest to other members in the research community." R2: "The idea of applying mutual information to understand KD and “dark knowledge” is very novel to the community." R3: "The connections between the information bottleneck theory and model distillation seem intriguing." R4: "The work is reasonably original. It is the first application of the IB principle to Knowledge Distillation (KD), yielding an observation that improves the state of KD."

Here, we wish to restate **the inspiration and potential implications of our work**. Several recent works [8,20,21,32] in the KD field have found a similar phenomenon: high performing teachers may not necessarily lead to better students. Some researchers guessed that the model capacity gap between strong teachers and weak students degrades knowledge transfer. However, they did not explain theoretically why gap exists and how gap affects KD. It has been troubling researchers of KD field.

Incidentally, we found that **a half-trained teacher gained distillation ability beyond that of a convergent teacher**. We then investigated the distillation performance of all teacher checkpoints and extended more model types, yielding consistent results. Further, we proposed and validated that **the snapshot ensemble distillation dramatically surpasses the widely-used full ensemble distillation**, which may change the state of ensemble distillation (**Contribution 1**).

We thought about why the optimal checkpoints tend to be in the middle, neither too early nor too late. Fortunately, the interpretation of deep networks by IB theory [26] gave us inspiration. They claimed that deep networks tend to obtain an efficient representation of the input, capturing the features relevant to the output and compressing those irrelevant. However, those features that are not relevant to the output (non-target category features) are exactly what KD needs. This prompted us to explore the connection between IB and KD. Formally, the objective function of KD usually includes two terms: the cross-entropy loss $L_{ce}$ (related with the category label $Y$) and the distillation loss $L_{dis}$ (related with the output of the teacher model). $L_{ce}$ contains complete information of the category label Y, which means sufficient $I(F; Y)$. $L_{dis}$ helps to improve students' performance because it contains information about input X, i.e., $I(X; F)$. Therefore, It is logical that using $I(X; F)$ to explain "dark knowledge". **From our view, the reason leading to the decline of KD ability of high-performing teachers may not be the model gap, but the excessive compression of non-target category information to the teacher models (i.e., $I(X; F)$).** Our proposed intermediate checkpoint distillation is a simple and effective way to avoid the excessive compression of $I(X; F)$. The link between IB and KD can help community researchers think further about the fundamentals of KD, analyse the unexplained phenomena in the past, and choose or design efficient teacher models in KD scenarios (**Contribution 2**).

A good distillation method means a good balance between $I(X; F)$ and $I(F; Y)$. Therefore, we proposed to search the optimal teacher checkpoints by IB curves. Honestly, the current algorithm is not satisfying to us. It considers the information entropy of the teachers but ignore the variation of the student structures, which can not ensure the optimal KD performance for all teacher-student pairs. In the next step, we will improve the algorithm and also hope that the followers can propose more general algorithms. In addition, we also empirically suggested that the half-way teacher model can suffice for KD in the case of limited computing resources, which can be viewed as a practical trick for KD applications. (**Contribution 3**)

Although our current research work is not perfect, we hope it will be helpful to the research community of KD.

---

### Meta-Review · Area_Chair_TuQH · 2022-08-29

**Recommendation:** Accept
**Confidence:** Less certain

**Metareview:**

While this paper has 4 accept recommendations among the four reviewers, I have serious misgivings about the content of this paper. The main experimental insight, that a less trained teacher sometimes performs better, is already known and unsurprising. In fact it's the very point of KD that makes that result interesting --- why should it be better to aim for a noisy target than the true target? On any dataset, if we train the teacher for long enough, we will eventually recover the exact labels to arbitrary precision. In that sense, all KD is with an "intermediate" trained model, and the only question is (has always been) just how early to stop. The next issue with the paper is that the authors claim to "explain" theoretically why KD works from the perspective of information bottleneck theory, however what they offer falls short. The “theory” is more like a story, with significant gaps. Most significantly, there is no logic to carry the leaps from stories of how mutual information evolves to why knowledge distillation should work. Moreover what the authors call "mutual information" in their experiments is not actually mutual information and the surrogates they use seem odd choices that are not consistent. For I(F;Y) the authors look at the output of the teacher model but for I(X;F) the authors look at an intermediate layer of F, training a decoder to predict X from the last convolutional layer of F. Why should the information contained in this middle layer of the teacher model matter when the student only accesses the teacher's output? My ambivalence with this paper is two-fold: (i) that the experimental findings are the main contribution and they are by themselves not sufficient for publication and (ii) that the IB component of this paper is misrepresented as a theoretical explanation of the efficacy of KD but actually it falls short.

Unfortunately I’m discovering these concerns and expressing them after the discussion, hence my recommendation to accept the paper on the basis of the reviewer's initial recommendations. If the work is accepted, I expect the authors to edit it responsibly to remove all misleading claims that suggest that the paper provides a propert theoretical account for why KD works (they certainly have not), versus a speculative intuition, and to be much more careful to disambiguate the quantities that they track from actual mutual information.

**Award:**

No

---

### Decision · Program_Chairs · 2022-09-14

Accept